# Does Environmental Credit Rating Promote Green Innovation in Enterprises? Evidence from Heavy Polluting Listed Companies in China

**DOI:** 10.3390/ijerph192013617

**Published:** 2022-10-20

**Authors:** Minxin Zuo, Tao Wu

**Affiliations:** School of Economics, Jiangxi University of Finance and Economics, Nanchang 330013, China

**Keywords:** environmental credit rating, green innovation, heterogeneous timing DID, China

## Abstract

Environmental credit rating (ECR) is a novel environmental governance tool proposed by China, but its implementation effect is still unknown. This study analyzed whether it achieves the goal of encouraging green innovation in enterprises. Based on the green patent data of listed companies in heavy polluting industries in China from 2010 to 2018, we constructed a heterogeneous timing difference-in-differences model to empirically study the impact of the ECR policy on green innovation. We find that the policy has significantly promoted heavy polluting enterprises’ green innovation. Moreover, the results passed a series of robustness tests. Importantly, we find that the policy has a positive effect on enterprises’ green innovation through the reputation mechanism and financing mechanism. Furthermore, the incentive effect of the policy varies with enterprise characteristics and regional characteristics: the green innovation effect of the policy is more obvious in large-sized and state-owned companies and companies in regions with low fiscal pressure and a high level of financial development are more likely to induce firms’ green innovation. Our research will be of practical value to China’s environmental management, as well as global value to other countries.

## 1. Introduction

Environmental degradation and resource depletion have become primary problems for the sustainable development of human society. After the long-term pursuit of high-speed economic growth at the expense of the environment and resources, China now faces the dual obstacles of environmental pollution and weak economic growth. In recent years, the Chinese government has gradually attached importance to green development. Green development is inseparable from green innovation, especially green technology innovation. Only by integrating green technology innovation into the production and manufacturing process can polluting enterprises reduce pollution emissions, thereby decreasing the negative impact of enterprise production activities on the ecological environment, compensating for the compliance costs caused by environmental regulations, and promoting environmental improvement and green transformation of enterprises. Therefore, it is of great practical significance to study how to promote firms’ green innovation.

Environmental regulation is an important tool for companies to adopt environmental initiatives [1]. The Chinese government has formulated a series of environmental laws and regulations and is increasingly implementing market-based environmental regulatory tools, such as sewage charges, pollutant discharge rights trading, and carbon emission trading. These tools have overcome many shortcomings of traditional command-type tools and have been successful to some extent, but also have some limitations. Therefore, policymakers and scholars continue to focus on how to formulate environmental regulations that are compatible with the development requirements of the times. 

At the end of 2013, four departments of the State Council of China jointly issued the “Administrative Measures for Enterprise Environmental Credit Rating (Trial Implementation)” (hereafter, the measures). After that, provinces have successively issued provincial-level normative documents around the relevant provisions of the measures and carried out ECR work for polluting enterprises. Based on imitating the credit rating system of the traditional financial market, the environmental credit rating (ECR) system has developed into a new type of environmental governance tool for China. In the existing literature, the jurisprudence of the system has been fully studied [2,3,4], but the effectiveness of the system has yet to be examined. Whether it will achieve its goal of promoting green innovation is also unknown.

However, research on environmental regulation and corporate green innovation has long been a hot topic in academic circles, and many scholars have carried out rich discussions on it, but no consensus has been reached. Porter and Van der Linde argued that sound environmental regulations could incentivize technological innovation to offset the negative impact of compliance costs [5]. Jaffe and Palmer proposed the “narrow Porter hypothesis” based on Porter’s theory, that is, flexible regulation produces a greater “incentive effect” for innovation [6]. Some scholars support the Porter hypothesis based on empirical research [7,8,9]. However, other scholars disagree. Some scholars have empirically studied the US manufacturing industry and found that the innovative effects of environmental regulation do not exist [10]. Some scholars argue that environmental regulations increase costs for businesses and weaken their motivation for technological innovation [11,12,13]. The abovementioned empirical studies have large differences in research objects, research samples, and research methods, making it difficult to reach a consensus on whether environmental regulation can produce an “innovation compensation effect”. In addition, most of the scholars focus on the investigation of the effects of regulatory policies, such as environmental laws and emission trading systems, while there are no studies on the policy effects of the ECR system. 

In this study, we regard the implementation of the ECR policy as a quasi-natural experiment to explore its impact on green innovation in enterprises. Then, first, we use the green patent data of heavy polluting listed companies in China’s A-share listed companies and establish a heterogeneous timing DID model based on the quasi-natural experiments. Next, we compare before and after policy implementation in pilot and non-pilot regions and examine the impact of the policy on green innovation in listed companies.

This study may make the following contributions. First, this study examines the relationship between environmental regulation and green innovation from the new perspective of ECR. Second, this study examines the policy effects of ECR for the first time by using empirical research methods. Finally, based on theoretical and empirical analysis, this study verifies the mechanism of ECR.

## 2. Literature Review and Hypothesis Development

### 2.1. Research on ECR

Credit rating can improve the efficiency of financial market operations by providing true and reliable public information through three major functions: information intermediary function, low-cost coordination mechanism, and quasi-regulator status [14]. ECR is a new ecological governance tool applied in the environmental field by imitating the credit rating model of the financial market [15]. The measures define the concept of ECR as follows: ECR refers to the environmental protection department, according to the information of the company’s environmental behavior, by the specified indicators, methods, and procedures, to assess the environmental behavior of enterprises. It carries out credit evaluation, determines credit rating, and discloses it to the public for public supervision and environmental management methods used by relevant departments, institutions, and organizations. The system integrates environmental protection departments, relevant administrative departments, financial institutions, and social organizations, and effectively combines command, market, contract, voluntary, and other governance tools, which is in line with the requirements of the theory of regulatory transformation on the diversification of governance subjects and regulatory tools and reflects strong market-oriented attributes [3].

Based on the information available, the ECR system appeared as a tool for environmental information disclosure in the early stage, and its action path resembles that of environmental information disclosure to a certain extent. Environmental credit disclosure is the “third wave” of environmental regulation after control-command and market [16], which exerts pressure on enterprises to force environmental protection and green innovation activities [17]. Some scholars have empirically studied China’s environmental disclosure policies, and they have found that strengthening environmental information disclosure reduces enterprises’ emissions, increases environmental protection investments, and optimizes the regional environment [18,19]. Wang and Wang further found that the implementation of policies aimed at improving the transparency of environmental information has significantly increased the enthusiasm for green innovation in enterprises with higher environmental risks [20].

In the above papers, scholars mainly focus on the institutional design and function introduction of ECR, and pay less attention to the application of ECR, while our research focuses on the application of ECR in enterprise management and environmental governance. Besides, the DID model has been widely used to estimate the effects of environmental information disclosure policies. Our research has adopted a similar but differentiated method, that is, the heterogeneous timing DID model (Staggered DID), which is beneficial for overcoming endogeneity and suitable for estimating the effect of the gradual implementation of a policy in the affected group.

### 2.2. ECR and Green Innovation

Existing studies have fully demonstrated that market-based environmental regulation tools can effectively guide the green innovation of enterprises by playing a market-oriented role [21,22,23]. The system of environmental credit evaluation also reflects the characteristics of being market-based. The measures stipulate that the results of ECR will be reported to the financial regulatory authorities, then, the green credit decisions of commercial banks can refer to the results, and insurance institutions can also adjust the insurance fee standard according to the results. In addition, relevant government departments can also provide tax incentives or penalties to relevant enterprises according to the results of ECR and provide preferential treatment or restrictions in government procurement activities. Driven by market factors, enterprises receiving ECR have a stronger incentive to adopt environmental protection strategies; strengthen green technology innovation to reduce pollution emissions; adhere to the commitment to environmental integrity; improve the ECR scores of enterprises, and obtain more economic resources, including government subsidies, tax relief, government procurement, and credit support, to enhance their competitiveness. Based on the above analysis, we propose Hypothesis 1.

**Hypothesis** **1.***The implementation of ECR system will promote green innovation in participating enterprises*.

### 2.3. The Impact Mechanism of ECR on Green Innovation

#### 2.3.1. Reputation Mechanism

Enterprise social trust, that is, enterprise reputation, has a greater impact on enterprise green innovation, enterprises with higher social trust are more favored in the financing, and sufficient funds will provide important support for enterprise green innovation [24]. Enterprises carrying out green innovation activities also send a positive signal to society that they attach importance to environmental protection, which is conducive to enterprises obtaining good social evaluation, thus helping enterprises improve their reputation [25]. The ECR system not only evaluates the environmental behavior of enterprises but also evaluates the environmental ethics, environmental attitude, and environmental behavior preparation of enterprises. From the original intention of system design, the ECR system relies on a reputation mechanism to work. The results of ECR will have a direct impact on the social reputation of enterprises, thus affecting the confidence of consumers and shareholders in related enterprises and producing incentive or deterrent effects on enterprises [4]. The environmental credit of enterprises is not only an important manifestation of their social integrity but also an important basis to judge whether they have a sense of social responsibility. Good ECR results help improve corporate reputation and its social image, and thus, enterprises take the initiative to assume higher environmental responsibility, increase investment in green technology research and development, and promote green innovation of enterprises out of the consideration of cherishing reputation. We, therefore, propose Hypothesis 2.

**Hypothesis** **2.**
*Corporate reputation is an effective mechanism for the ECR system to affect corporate green innovation.*


#### 2.3.2. Financing Mechanism

Enterprise innovation activities require a large amount of capital investment, and sufficient funds are the basic guarantee for enterprise green innovation. The existing research shows that listed companies disclose more environmental information, which is conducive to reducing the degree of information asymmetry between banks and enterprises, reducing the risk of credit mismatch, improving the availability of credit, reducing credit costs, and alleviating the financing constraints of companies [26,27,28]. In addition, high-quality environmental information will affect investors’ cognition, improve investors’ stock expectations and investment confidence in listed companies, and help to clear the financing obstacles of listed companies in the financing capital market [29]. Environmental information produced an information increment effect, which made the external stakeholders of listed companies improve their evaluation of the company and at the same time reduce the investment return requirements for listed companies, which will reduce the cost of equity financing of listed companies [30]. The disclosure of ECR results is an important part of the operation of the ECR system. The ECR results are evaluated and graded by the environmental assessment department according to many indicators in many aspects, such as corporate responsibility, environmental violations, and environmental management. Finally, the environmental protection department makes official publicity, and the enterprise’s environmental information transmitted is extremely authoritative, scientific, and comprehensive. To a large extent, this solves the problem of information asymmetry between participating enterprises and financial institutions, investors, and consumers, which is conducive to improving the availability of enterprise financing, reducing the cost of enterprise financing, alleviating the constraints of enterprise financing, and thus promoting the green innovation of enterprises. Hence, we propose Hypothesis 3 as follows.

**Hypothesis** **3.***The ECR system supports green innovation by easing corporate financing constraints*.

## 3. Research Design

### 3.1. Samples and Data 

As the heavy polluting enterprises are the common key evaluation objects in the environmental credit evaluation of various regions, we select listed A-share companies in heavy polluting industries from 2010 to 2018 as the research object. According to the 2012 CSRC industry classification standard, we identify B06, B07, B08, B09, B10, C15, C17, C18, C19, C22, C25, C26, C27, C28, C29, C30, C31, C32, and D14 as heavy polluting industries. Then, we perform the following processing on the sample data: eliminate the samples with financial or other abnormalities (including ST, *ST listed companies) and delete the listed companies with more missing research variable data. Then, we finally obtain balanced panel data of 540 listed companies for nine consecutive years, with a total of 4860 observations. To prevent the influence of extreme outliers, the main continuous variables are winsorized at 1% and 99%.

### 3.2. Variables and Model 

#### 3.2.1. Variable Introduction

Green innovation (*GI*). Scholars usually use innovation input or output to measure the innovation level, but it is difficult to distinguish green innovation from non-green innovation. In recent years, the method of measuring the green innovation activities of enterprises by the number of green patents has become more and more popular among scholars in the field of environmental economics [13,31,32]. This is because the source of patent data is authoritative, and green patents are identified by the relevant regulations of the World Intellectual Property Organization (WIPO), which ensures the accuracy and availability of the data. In addition, green patents are relatively weakly affected by unobservable factors [33]. Patent applications reflect the innovation progress of enterprises and are likely to impact enterprises during the application period. Patent application data are more reliable, timely, and stable than patent authorization data [34]. Therefore, this study uses the number of green patent applications to measure the green innovation behavior of enterprises, represented by ${GI}_{i,p,t}$, which is the natural logarithm of the sum of 1 and the number of green patent applications in year *t* by enterprise *i* in region *p*.

Environmental credit rating (*ECR*). This is the core explanatory variable of this study. It is used to explain whether listed companies have accepted the ECR. For enterprises that have implemented the ECR system in their regions from 2010 to 2018, the value is 0 before the implementation of the system and 1 after the implementation. For enterprises in regions that have not implemented the system, the value is always 0. Considering that the ECR work was carried out from the first half of the year to the end of the year, and it also takes time for enterprises to adjust under the influence of the policy, we conduct the following treatment on the implementation time of the policy. If a region issues the ECR policy in the first half of the year (before June 30), it is deemed that the policy has been implemented in the year. If a region issues the policy in the second half of the year (after June 30), it is deemed that the policy will be implemented in the next year. In the end, a total of five policy implementation years and 20 provincial-level administrative units implemented the environmental credit evaluation system, namely, in 2014 (Guizhou Province and Yunnan Province), 2015 (Guangdong Province, Hebei Province, Hunan Province, Inner Mongolia Autonomous Region, Tibet Autonomous Region, and Sichuan Province), 2016 (Shaanxi Province, Liaoning Province, and Henan Province), 2017 (Fujian Province, Ningxia Autonomous Region, Anhui Province, Hubei Province, and Shandong Province), and 2018 (Jiangxi Province, Chongqing City, Heilongjiang Province, and Jilin Province).

Referring to the existing research results [13,32,35], we select the relevant characteristic variables at the enterprise level that may affect green innovation as the control variables of the model. These variables include the size of the enterprise (*Size*), which is the natural logarithm of the company’s total assets at the end of the year; the age of the enterprise (*Age*); the shareholding structure (*First*), which refers to the shareholding ratio of the largest shareholder; financial leverage (*Lev*), measured by the asset–liability ratio; growth capability (*Growth*), measured by the growth rate of operating yield; profitability (*Roa*), measured by return on equity; cash flow (*Cash*), measured by the cash recovery rate and board size (*Board*), measured as the natural logarithm of the board size. The descriptive statistics of the main variables are shown in Table 1. The mean value of ${GI}_{i,p,t}$ is 0.28, with a minimum value of 0 and a maximum value of 6.874, which indicates that the level of green innovation of Chinese heavy polluting listed enterprises is uneven and there are large differences in the sample.

#### 3.2.2. Model Specification

Since the policy time for the implementation of the ECR system in various provinces is not uniform, to study the impact of the system on green innovation, this study uses the heterogeneous timing DID method, which is very mature in the field of policy evaluation [36,37,38]. We construct Equation (1) as follows.
(1)$${GI}_{i,p,t}=\alpha +\beta {ECR}_{i,p,t}+\gamma CVs_{i,p,t}+\delta_{p}+\varphi_{t}+\varepsilon_{i,p,t}$$
where *i* represents the listed company, *p* represents the region where the company is located, *t* represents time, and $\varepsilon_{i,p,t}$ represents the random disturbances. ${ECR}_{i,p,t}$ is the core explanatory variable of the model and the main variable of interest. It refers to the ECR, that is, the region *p* where company *i* is located starts or has implemented the ECR system in year *t*; it is either 1 or 0. $CVs_{i,p,t}$ represents control variables.

β, the regression coefficient of ${ECR}_{i,p,t}$, is also the focus of our attention. It reflects the degree of influence of ECR on green innovation in enterprises. If the regression coefficient is positive, it indicates that the green innovation effect of heavy polluting enterprises affected by the ECR is significant compared to other heavy polluting companies. Finally, the model also introduces the regional fixed effect $\delta_{p}$ to control the unobservable factors at the regional level that may affect the green innovation of enterprises; the time fixed effect $\varphi_{t}$ is introduced to control the unobservable factors that change with time but not with individuals, to improve the effectiveness of the causal analysis of the policy evaluation.

## 4. Empirical Analysis

### 4.1. Benchmark Regression Analysis

#### 4.1.1. Benchmark Regression Results

The regression results of the impact of ECR on green innovation in heavy polluting enterprises are shown in Table 2. According to Equation (1), we regress two cases: without adding control variables and adding control variables while controlling for both time-fixed effects and region-fixed effects. The results are shown in columns (1) and (2), respectively. The estimated coefficients of the core explanatory variables ${ECR}_{i,p,t}$ are all significantly greater than 0 at the 5% level. Most importantly, column (2) shows the regression results of the benchmark model; the coefficient of ${ECR}_{i,p,t}$ is 0.0598 and is significant at the 5% level. This is sufficient to show that the implementation of ECR has a significant incentive effect on the green innovation activities of participating heavy polluting enterprises, which confirms Hypothesis 1. 

#### 4.1.2. Parallel Trend Test and Dynamic Effect Analysis

Using the event study method proposed by Jacobson and Sullivan [39] and the research framework of McGavock [40] and Wu et al. (2021) [41], this study verifies the parallel trend hypothesis of the DID model and explores the dynamic effects of the implementation of ECR policy. We design the model as follows.
(2)$${GI}_{i,p,t}=\delta_{0}+\overset{6}{\underset{n=1}{\sum }}\beta_{pren}D_{pren}+\beta_{current}D_{current}+\overset{4}{\underset{n=1}{\sum }}\beta_{aftern}D_{aftern}+\gamma CVs_{i,p,t}+\delta_{p}+\varphi_{t}+\varepsilon_{i,p,t}$$
where $D_{pren}$, $D_{current}$, and $D_{aftern}$ denote the interaction terms of the policy time dummy variables before, in the current year of implementation, and after the implementation of the ECR policy, respectively, with the policy dummy variables. We take the first 7 years of the implementation of the policy as the comparison benchmark, and thus, $D_{pre7}$ is not added to the model. The coefficients corresponding to the three dummy variables are $\beta_{pren}$, $\beta_{current}$, and $\beta_{aftern}$, and the meanings of the remaining variables and symbols are consistent with Equation (1). 

To reflect the dynamic policy effect of ECR more intuitively, this study plots the regression results of Equation (2) as a graph. As shown in Figure 1, before the implementation of the ECR policy, the corresponding coefficients are not significant, and changed from negative to positive slowly, which shows the parallel trend assumption is satisfied.

As we can also see in Figure 1, $\beta_{current}$ and $\beta_{aftern}$ are greater than 0, indicating that after the implementation of ECR, the green innovation level of heavy polluting enterprises has improved. However, $\beta_{current}$ and $\beta_{after1}$ are not significant until the second year after the implementation of the policy, when the policy has a significant green innovation incentive effect; this shows that there is a short-term lag effect of the policy effect. The reason may be that the ECR work lasts a long time. After the environmental assessment department completes the rating of the enterprise, it is also necessary to publicize the results on the government website and inform third parties, such as financial institutions. The feedback of financial institutions and relevant government departments to relevant enterprises might not be timely. Moreover, shareholders, consumers and other stakeholders usually obtain the results after the announcement, and thus, the policy has a certain lag effect.

### 4.2. Robustness Check

#### 4.2.1. Placebo Test

For the regression results to be more convincing, a placebo test is essential. We select an indirect placebo test method, that is, a non-parametric displacement test, which has been widely used by scholars [42,43,44]. Specifically, we randomly sample the interaction terms of all policy dummies and time dummies 500 times and perform the regression according to Equation (1), obtaining 500 differential regression coefficients of DID. Then, we plot the regression coefficients of ${ECR}_{i,p,t}$ and kernel density. 

As shown in Figure 2, the regression coefficients are normally distributed, and the estimated coefficient of ${ECR}_{i,p,t}$ in benchmark model Equation (1) is 0.0598; that is, the dotted line on the right-hand side is located at the low tail position of the placebo test coefficient distribution. This shows that the placebo test has been passed.

#### 4.2.2. Counterfactual Test

We also design a counterfactual test, that is, we conduct a study on non-heavy polluting companies that are not the key objects of the ECR policy [45]. Specifically, we sample 1213 listed companies from 2010 to 2018 that were not part of heavy polluting industries, and the variables and models remained unchanged. The regression results are shown in Column (1) of Table 3. The estimated coefficient of ${ECR}_{i,p,t}$ is negative and not significant, which shows that our conclusion is reliable.

#### 4.2.3. Change How Green Innovation Is Measured

This study re-evaluates the interpreted variable ${GI}_{i,p,t}$, and uses the natural logarithm of the number of green patent grants of listed companies and the sum of 1 as the measurement standard for it [35]. As shown in column (2) in Table 3, the regression coefficient of ${ECR}_{i,p,t}$ is significantly positive, which is consistent with the conclusions of the previous regression.

#### 4.2.4. Delete the Disputed Samples

A handful of regions have tried to evaluate the environmental behavior of some key polluting enterprises before 2014. Although the nature of such work is substantially different from the content of the ECR policy, it may also confuse the policy effect. Therefore, the samples of enterprises belonging to three regions in Chongqing City, Jiangsu Province, and Hubei Province were eliminated, because these three regions issued local documents relevant to the conducted audit for enterprises’ environmental behavior in 2011, 2012, and 2013. Then the regression test is carried out. The results are shown in column (3) of Table 3. The estimated coefficient of ${ECR}_{i,p,t}$ is still positive and passed the significance test at the 5% level, indicating that the benchmark regression results are robust.

#### 4.2.5. Consider Possible Missing Variables

In the previous analysis, fixed effects of time and region are added to the benchmark regression model to control the influence of some unobservable factors. However, some unobservable enterprise-level variables are inevitably missed, which may impact green innovation. Therefore, based on the benchmark model, we further control fixed effects at the enterprise level. The regression results are shown in column (4) in Table 3. The coefficient of ${ECR}_{i,p,t}$ is significantly positive, which strengthens the robustness of the benchmark regression results.

### 4.3. Mechanism Analysis

The results of the benchmark regression analysis show that the ECR policy promotes the green innovation of heavy polluting enterprises. To verify the previous hypotheses, this study further tests the two possible mechanisms of corporate reputation and financing constraints.

#### 4.3.1. Reputation Mechanism

To verify the mechanism of corporate reputation, this study constructs the interaction term of corporate reputation and ECR and adds it to the benchmark regression model. The specific model is as follows.
(3)$${GI}_{i,p,t}=\alpha +\beta_{0}{ECR\;}_{i,p,t}\times Repu_{i,p,t}+\beta_{1}Repu_{i,p,t}+\beta_{2}{ECR\;}_{i,p,t}+\gamma CVs_{i,p,t}+\sigma_{i}+\delta_{p}+\varphi_{t}+\varepsilon_{i,p,t}$$

In Equation (3), the variable $Repu_{i,p,t}$ represents corporate reputation. In this study, we take the natural logarithm of corporate intangible assets as a proxy variable for corporate reputation [46]; the meanings of other variables and indicators are the same as in Equation (1). Additionally, we further control for the effect of firm heterogeneity $\sigma_{i}$. The estimated coefficient $\beta_{0}$ is the core of this model. If $\beta_{0}$ is significantly greater than 0, it means that ECR promotes corporate green innovation by affecting corporate reputation. 

The regression results are shown in column (1) of Table 4. The coefficient of the interaction term is significantly positive at the 1% level, indicating that corporate reputation is an important path for an environmental credit evaluation to affect corporate green innovation. The conclusions drawn are consistent with Hypothesis 2. The ECR system simplifies complex corporate environmental information and provides the public with more efficient information [15], which is conducive to deepening society’s understanding of enterprises. Under the influence of ECR policy, enterprises have a strong incentive to carry out green technology innovation to obtain a good reputation. 

#### 4.3.2. Financing Mechanism

Similarly, to verify whether the environmental credit evaluation affects the green innovation activities of enterprises through the financing mechanism, we add the interaction term of financing constraints and ECR to the benchmark regression model and construct the following model.
(4)$${GI}_{i,p,t}=\alpha +\beta_{0}{ECR\;}_{i,p,t}\times Fins_{i,p,t}+\beta_{1}Fins_{i,p,t}+\beta_{2}{ECR\;}_{i,p,t}+\gamma CVs_{i,p,t}+\sigma_{i}+\delta_{p}+\varphi_{t}+\varepsilon_{i,p,t}$$

In Equation (4), the variable $Fins_{i,p,t}$ represents the corporate financing constraint, which is measured by the credit availability of listed companies [47]. Its specific meaning is the ratio of the total long-term borrowings of the company to the total assets at the end of the period. A smaller value of $Fins_{i,p,t}$ means the enterprise faces smaller financing constraints. Other variables in this model are consistent with Equation (1). The estimated coefficient $\beta_{0}$ of ${ECR}_{i,p,t}\times Fins_{i,p,t}$ is our top concern in this model; it is used as an important foundation to verify whether the financing mechanism holds. 

The regression results are shown in column (2) of Table 4. $\beta_{0}$ is 0.236, and passes the significance test at the 5% level, which means that the ECR policy provides an impetus for the green research and development activities of enterprises by easing their financing constraints. This confirms Hypothesis 3. One of the design purposes of the ECR system is to better integrate environmental credit with green financial policy. Therefore, the system depends to a large extent on the cooperation of financial institutions and other market players. The disclosure of the ECR results effectively reduces the degree of information asymmetry between enterprises and financial institutions, such as banks and insurance companies. The cost of bank credit decision-making is reduced, and the availability of credit for enterprises is improved, thereby supporting green innovation in enterprises.

## 5. Heterogeneity Analysis

### 5.1. Heterogeneity Effects of Enterprise Characteristics

#### 5.1.1. Heterogeneity of Size 

Due to the large differences in the development ability, capital ability, and technical level of enterprises of different sizes, the degree of influence of enterprises’ green innovation behavior by environmental policies is also different [48]. To explore whether the relationship between the ECR system and enterprise green innovation is influenced by firm size, this study divides the full sample into large-sized and small-sized company subsamples. Following Deng et al. [48], if the value of the natural logarithm of the total assets of a company is higher than the median of sample firms, we regard it as large-sized, otherwise it is small-sized. Then, the two sets of samples are regressed. 

As shown in Table 5, the estimation coefficients of ${ECR}_{i,p,t}$ in the large-sized samples are significantly positive, but not significant in the small-sized group. This shows that the ECR system has effectively encouraged large-sized firms to carry out green innovation activities, but has had no significant incentive effect on small-sized firms. 

#### 5.1.2. Heterogeneity of Ownership

The green innovation behavior of firms is different due to the different properties [13]. To examine whether the relationship between the ECR system and enterprise green innovation is influenced by firms’ property, we divide the original samples into state-owned and non-state-owned samples. Then, we conduct regressions according to the benchmark regression model.

As Table 5 shows, the implementation of the system has had a significant positive impact on state-owned firms’ green innovation, but no significant impact on non-state-owned firms. Non-state-owned heavy polluting firms are weaker than state-owned firms, have insufficient funds, and lack competitiveness, making it difficult to carry out green innovation activities. 

### 5.2. Heterogeneity Effects of Regional Characteristics

#### 5.2.1. Heterogeneity of Different Financial Development Levels

The previous analysis shows that the system affects the heavy polluting firms’ green innovation through the financing mechanism. Due to the different levels of financial development, there is a large difference in green credit efficiency among regions. To examine the effect of this difference on policy effectiveness, we construct the regional financial development level index by using the ratio of the sum of total deposits and loans of banking financial institutions in various regions to the GDP of the region [49]. Then we divide the sample into high and low financial development level samples. The benchmark regression model is used for the regression, and the results are shown in columns 1 and 2 of Table 6.

We find that the policy significantly promotes green innovation for heavy polluting firms in regions with high financial development levels, and the incentive effect for green innovation is insignificant in regions with low financial development levels. The reason may be that financial institutions, such as banks and insurance companies affiliated with areas with high levels of financial development, have greater demand for capital lending than those affiliated with areas with low levels of financial development, and the policy has reduced the degree of information asymmetry between banks and firms, improving corporate credit accessibility, and thus supporting polluting enterprises to carry out green technology research and development.

#### 5.2.2. Heterogeneity of Different Fiscal Pressure Levels

The increased fiscal pressure may reduce the government’s environmental governance efficiency [50]. Since the ECR is carried out by each local government, the policy effect may affect by the local government’s fiscal pressure. Therefore, this study constructs an indicator of fiscal pressure level. Similarly, according to the median of the value of fiscal pressure, if a region faces a higher fiscal pressure than the median value, it is a high fiscal pressure region, otherwise, it is a low fiscal pressure region. Then we divide the study sample into high and low fiscal pressure level samples. The regression results are shown in the third and fourth columns of Table 6.

The results indicate that the policy significantly promotes the green innovation of the heavy polluting enterprises in regions with less fiscal pressure, while the green innovation effect is not obvious in regions with high fiscal pressure. The reason may be that stable financial resources are important guarantees for a government to conduct environmental governance [51]. 

## 6. Conclusions

Based on the green patent data of listed firms in heavy pollution industries from 2010 to 2018 in China’s A-share listed companies, this study uses the implementation of the ECR system as a quasi-natural experiment and then constructs a heterogeneous timing DID model to empirically analyze the effect of the policy on enterprises’ green innovation. The findings show that the policy exerts a significant effect on green innovation in heavy pollution firms, which demonstrates the existence of Porter’s hypothesis in China. First, the policy significantly promotes green innovation in heavy pollution firms, but there is a short-term lag in this innovation incentive effect. Second, the policy exerts a positive effect on firms’ green innovation through the reputation mechanism and financing mechanisms. Third, the policy-generated incentive effect for green innovation varies by firm and regional characteristics: the green innovation effect is more pronounced among large-sized and state-owned firms, and more likely induced in regions with low fiscal pressure and a high financial development level.

The conclusions of this study reveal the green innovation incentive effect of ECR policy, and theoretically summarize and enrich the related research on green innovation and environmental governance. Based on this, we propose the following policy suggestions. First, professional construction of the ECR system should be enhanced, and the efficiency of the operation of the system should be improved, thereby alleviating the lag of the policy effect. Second, the government can accelerate the establishment of a unified rating system for environmental credit, while simultaneously introducing third-party monitoring mechanisms to ensure that the rating process is open and transparent, to promote equitable law enforcement by all environmental rating agencies, to eliminate problems of law enforcement discrimination, rent seeking, and to narrow the gap in policy effectiveness due to corporate and regional differences. Third, the government should optimize the financing mechanism of enterprises’ green innovation, effectively link the green credit policy and ECR system, and play a leading role in constructing the market mechanism for green innovation.

There are still several possible expansions for this study: First, if one can obtain the environmental credit score of each enterprise, the impact of ECR on enterprise green innovation can be revealed more intuitively; second, our data are limited to listed companies, if data of non-listed companies are available in the future, our analysis and conclusions will be more robust. Further research can focus on these possibilities and how ECR impacts firms’ environmental protection investment.

## Figures and Tables

**Figure 1 ijerph-19-13617-f001:**
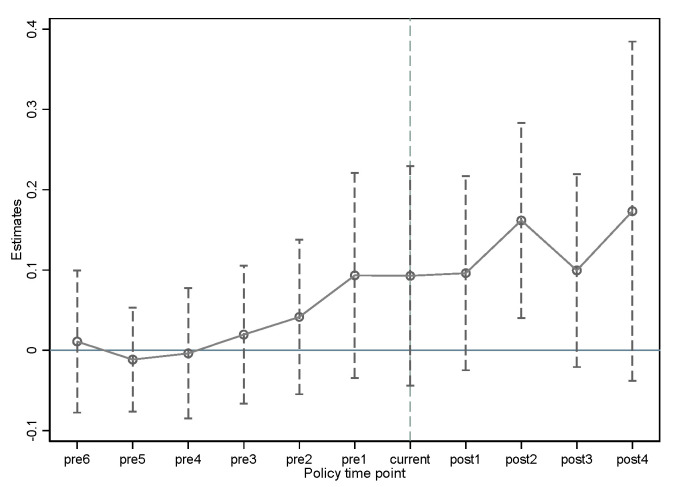
Parallel trend and dynamic effect.

**Figure 2 ijerph-19-13617-f002:**
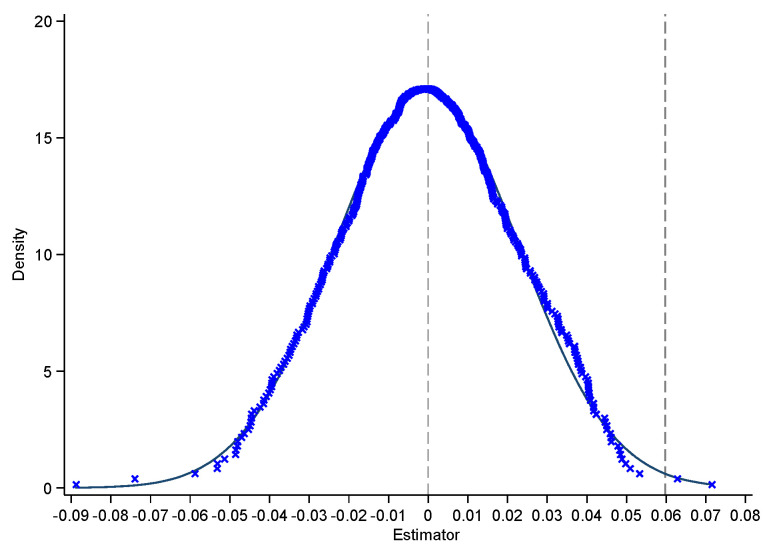
Placebo test.

**Table 1 ijerph-19-13617-t001:** Descriptive statistics for primary variables.

Variables	N	Mean	Sd	Min	Max
${GI}_{i,p,t}$	4860	0.280	0.724	0.000	6.874
*Size*	4860	22.308	1.308	18.158	28.520
*Age*	4860	2.741	0.388	0.000	3.738
*Growth*	4860	0.177	0.433	−0.569	3.261
*Lev*	4860	0.445	0.228	0.007	2.992
*Roa*	4860	0.042	0.135	−1.038	7.445
*First*	4860	26.698	18.210	0.414	69.970
*Cash*	4860	0.054	0.071	−0.177	0.243
*Board*	4860	2.175	0.192	1.609	2.708

**Table 2 ijerph-19-13617-t002:** Regression results.

Variables	${GI}_{i,p,t}$	
	(1)	(2)
${ECR}_{i,p,t}$	0.0583 ** (0.0225)	0.0598 ** (0.0227)
*Size*		0.214 *** (0.0658)
*Age*		−0.141 ** (0.0541)
*Growth*		−0.0388 ** (0.0188)
*Lev*		−0.0645 (0.147)
*Roa*		−0.107 (0.107)
*First*		0.00305 ** (0.00149)
*Cash*		0.295 ** (0.141)
*Board*		0.142 (0.116)
*Year-FE*	Yes	Yes
*Province-FE*	Yes	Yes
*N*	4860	4860
*R* ^2^	0.066	0.227

Note: Robust standard errors clustered in region level are given in parentheses, *, **, and *** denote 10%, 5%, and 1% confidence.

**Table 3 ijerph-19-13617-t003:** Robustness check.

Variables	${GI}_{i,p,t}$			
	(1)	(2)	(3)	(4)
${ECR}_{i,p,t}$	−0.0050 (0.0277)	0.0452 * (0.0226)	0.0599 ** (0.0253)	0.0580 ** (0.0224)
*CVs*	Yes	Yes	Yes	Yes
*Year-FE/Province-FE*	Yes	Yes	Yes	Yes
*Firm-FE*	No	No	No	Yes
*N*	10913	4446	4257	4860
*R* ^2^	0.072	0.163	0.236	0.765

Note: The control variables are consistent with the benchmark regression model Equation (1). Due to space limitations, the results are not reported here. Robust standard errors clustered in region level 345 are given in parentheses, *, **, and *** denote 10%, 5%, and 1% confidence.

**Table 4 ijerph-19-13617-t004:** Regression results of mechanism test.

Variables	${\mathrm{GI}}_{i,p,t}$	
	Reputation Mechanism(1)	Financing Mechanism(2)
${ECR}_{i,p,t}\times Repu_{i,p,t}$	0.0384 *** (0.0130)	
${ECR}_{i,p,t}\times Fins_{i,p,t}$		0.236 ** (0.107)
*Year-FE*	Yes	Yes
*Province-FE*	Yes	Yes
*Firm-FE*	Yes	Yes
*N*	4815	4860
*R* ^2^	0.770	0.765

Note: The control variables are consistent with the benchmark regression model, and $Reput_{i,p,t}$, $Fins_{i,p,t}$, and $ECR_{i,p,t}$ are controlled. The results are not reported owing to space limitations. Robust standard errors clustered in region level 345 are given in parentheses, *, **, and *** denote 10%, 5%, and 1% confidence.

**Table 5 ijerph-19-13617-t005:** Heterogeneity effects of enterprise characteristics.

Variables	${\mathrm{GI}}_{i,p,t}$			
	Large-Sized Samples	Small-Sized Samples	State-Owned Samples	No State-Owned Samples
${ECR}_{i,p,t}$	0.116 ** (0.0475)	0.0318 (0.0306)	0.0756 * (0.0445)	0.0291 (0.0249)
*CVs*	Yes	Yes	Yes	Yes
*Year-FE/* *Province-FE*	Yes	Yes	Yes	Yes
*N*	2430	2430	2385	2475
*R* ^2^	0.332	0.060	0.289	0.072

Note: The control variables are consistent with the benchmark regression model. Robust standard errors clustered in region level 345 are given in parentheses, *, **, and *** denote 10%, 5%, and 1% confidence.

**Table 6 ijerph-19-13617-t006:** Heterogeneity effects of region characteristics.

Variables	${\mathrm{GI}}_{i,p,t}$			
	High Financial Development Level	Low Financial Development Level	High Fiscal Pressure Level	Low Fiscal Pressure Level
${ECR}_{i,p,t}$	0.0600 ** (0.0281)	0.0473 (0.0324)	0.0506 (0.0420)	0.0914 ** (0.0389)
*CVs*	Yes	Yes	Yes	Yes
*Year-FE/* *Province-FE*	Yes	Yes	Yes	Yes
*N*	2502	2358	2515	2345
*R* ^2^	0.287	0.149	0.172	0.304

Note: The control variables are consistent with the benchmark regression model. Robust standard errors clustered in region level are given in parentheses, *, **, and *** denote 10%, 5%, and 1% confidence.

## Data Availability

We compiled the policy implementation time manually from various local government websites and the Peking University Magical Law Database and gain manually the green patent data from CNIPA. All financial data and other enterprise characteristic data of listed companies in this paper are from WIND and CSMAR database. The data presented in this study are available on request from the corresponding author. The data are not publicly available due to privacy.

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
