# Peer review of "Does Environmental Credit Rating Promote Green Innovation in Enterprises? Evidence from Heavy Polluting Listed Companies in China"

_ijerph, 2022, doi:10.3390/ijerph192013617_

Round 1

Reviewer 1 Report

The article is well designed and informative. Its content is relevant for the profile of the journal. A few minor comments are presented in the file attached. 

Reviewer 2 Report

In the paper, This study analyzed whether ECR achieved the goal of encouraging green innovation among firms. A heterogeneous time-difference model was constructed to empirically investigate the impact of the ECR policy on green innovation based on green patent data of Chinese listed companies in the heavy pollution industry from 2010 to 2018. The policy was found to significantly promote green innovation among heavy polluting firms.

However, there has also been some general structural error. Therefore, the author(s) proposed that the document be thoroughly reviewed before it was submitted to the journal. I encourage the writers and wish them success in publishing.

a) I suggest that the author make appropriate changes to the abstract, keeping in mind that most readers will read your full paper if they are interested in what you write in the abstract. Therefore it needs to be attractive.

b) Moving towards the 'Introduction' part, it is too redundant, which seems a bit annoying, so please to streamline it. For example, lines 51-67 and 69-87 make me feel that they are two completely separate parts and the author should make them properly connected. In the final part of the introduction, the author should highlight more the innovative nature of the article's research, rather than seeming to introduce the individual sections.   

c) The literature review section, which the authors have divided into two sections according to categories, is fine, but the lack of discussion related to the technical level of the study or the methodology can make it difficult for the non-specialist reader to understand the paper. For example, an introduction to the heterogeneous timing DID model. In the “Research on ECR” section, the authors should have presented more about the application of ECR rather than just the functions of ECR. Also, the authors should include more literature comparisons to compare where your study differs from these papers in the analysis of the results.

d) The “additional analysis” in section 5 is not good as a chapter title, please replace it.

e) In line 289, the author is asked to consider the case of adding control variables while the control area is fixed.

f) There is extensive English revision required.
